# Defective IgG Class Switching in the Spleen of TRAF5-Deficient Mice Reveals a Role for TRAF5 in CD40-Mediated B Cell Responses During Obesity-Associated Inflammation

**DOI:** 10.3390/ijms26199494

**Published:** 2025-09-28

**Authors:** Tomomi Wakaizumi, Mari Hikosaka-Kuniishi, Yusuke Ozawa, Ayaka Sato, Chieri Iwata, Tsutomu Wada, Toshiyasu Sasaoka, Masashi Morita, Takanori So

**Affiliations:** 1Laboratory of Molecular Cell Biology, Graduate School of Medicine and Pharmaceutical Sciences, University of Toyama, 2630 Sugitani, Toyama 930-0194, Japan; 2Department of Clinical Pharmacology, University of Toyama, Toyama 930-0194, Japan

**Keywords:** antibody, B cell, high-fat diet, lymphocyte, obesity, CD40, spleen, TNF, TNFR, TRAF

## Abstract

Tumor necrosis factor receptor-associated factors (TRAFs) are a family of adaptor proteins that transmit signals from immunoregulatory receptors—such as TNF receptors, Toll-like receptors, and interleukin receptors—to coordinate immune and inflammatory responses. Among them, TRAF5 is highly expressed in lymphocytes and implicated in obesity-associated inflammation, but its role in secondary lymphoid organs during chronic low-grade inflammation remains unclear. We examined splenic B and T cell phenotypes in wild-type (WT) and *Traf5*-deficient (KO) mice fed a high-fat diet (HFD). Although lymphocyte composition was broadly comparable, KO mice showed reduced spontaneous immunoglobulin G2c (IgG2c) production ex vivo—about 1.5-fold lower than WT. Notably, despite elevated TNF-α and CD40 ligand (CD40L) expression in HFD-fed KO splenocytes, IgG2c production remained diminished—about 1.9-fold lower than WT—upon soluble CD40L stimulation, indicating impaired CD40-mediated class-switch recombination (CSR). Consistently, B cells from KO mice on a normal diet exhibited reduced activation-induced cytidine deaminase (AID) expression—about 4.4-fold lower than WT—after CD40L stimulation, and decreased IgG2c secretion—about 6.6-fold lower—upon CD40L and IFN-γ co-stimulation in vitro. Collectively, these findings suggest that TRAF5 is involved in CD40-dependent CSR in B cells under inflammatory conditions and may contribute to sustaining adaptive immune responses during obesity-associated chronic inflammation.

## 1. Introduction

Tumor necrosis factor receptor-associated factors (TRAFs) function as key mediators that link receptors to intracellular signaling molecules, playing essential roles in cell development, survival, death, and inflammation under both physiological and pathological conditions [1]. TRAF5, one of the seven mammalian TRAF family members, is an adaptor protein that regulates signaling pathways mediated not only by canonical tumor necrosis factor receptors (TNFRs) but also by non-canonical Toll-like receptors (TLRs) and type I cytokine receptors for interleukins [2,3,4,5].

TRAF5 has been demonstrated to act as a potential regulator of TNFR superfamily members [6,7,8,9,10,11,12,13,14,15,16,17,18,19,20]. We previously demonstrated that TRAF5 plays a crucial role in CD40-induced antibody production and germinal center formation during the primary immune response [21]. TRAF5 is expressed at relatively high levels in B and T lymphocytes; thus, it is considered to play a pivotal role in lymphocyte responses and adaptive immunity. In mice, the absence of *Traf5* promotes pathological responses that contribute to autoimmune and inflammatory diseases [20,22,23,24,25,26,27,28]. In humans, genetic mutations in *TRAF5* have been associated with the development of autoimmune disorders, including rheumatic diseases [29,30].

Lipid metabolism plays a pivotal role in regulating immune homeostasis. The spleen—a major secondary lymphoid organ enriched with B and T lymphocytes—facilitates the production of pro-inflammatory cytokines and promotes the egress and recruitment of pathogenic lymphocytes to target organs, such as the heart and vasculature, adipose tissue, and liver, in response to metabolic abnormalities associated with cardiovascular diseases [31,32]. Disturbances in lipid metabolism enrich with fatty acids and cholesterol derivatives elevates the levels of key pro-inflammatory cytokines, including tumor necrosis factor-α (TNF-α), CD40 ligand (CD40L), and interferon-γ (IFN-γ, thereby promoting chronic low-grade inflammation [33,34,35,36,37,38,39]. Within this pro-inflammatory cytokine milieu, effector T cells—such as T helper 17 (Th17) and Th1 cells—and their associated cytokines aggravate obesity-related pathological responses [34,37,40,41,42]. Additionally, immunoglobulin G (IgG) antibodies produced through T-B cell interactions greatly contribute to insulin resistance [43].

Obesity is a major risk factor for cardiovascular diseases and related conditions, including hypertension and stroke. It also strongly associated with metabolic disorders such as fatty liver disease and prediabetes [44,45,46,47]. Dysregulation of lipid metabolism is a central driver of obesity-associated cardiovascular pathology. Aberrant lipid storage and elevated free fatty acids promote ectopic fat accumulation, particularly in the liver and heart, where these lipids activate immune receptors such as TLR4 and trigger NLRP3 inflammasome activation [48,49]. This sterile inflammation remodels adipose-tissue immunity by promoting macrophage infiltration, reducing the number and/or function of regulatory T cells (Tregs), and upregulating proinflammatory mediators such as TNF-α, CD40L, and IFN-γ [33,34,35,36,37,38,39,50,51,52]. In parallel, altered adipokine signaling—characterized by increased leptin and decreased adiponectin—together with insulin resistance further disrupts vascular and metabolic homeostasis [53,54,55]. Immune-cell-mediated disturbance contribute to hypertension via sympathetic and renal mechanisms, promote atherosclerosis and endothelial dysfunction, and drive tissue-specific remodeling processes, including cardiac remodeling and metabolic dysfunction-associated steatotic liver disease (MASLD) and its progressive form, metabolic dysfunction-associated steatohepatitis (MASH) [45,56,57,58,59,60]. Clinically, these processes manifest as increased risk of ischemic stroke, heart failure, and other obesity-related comorbidities [47,61,62]. Notably, the link between cardiovascular and autoimmune diseases has garnered increasing attention in recent years [63]. Therefore, elucidating the impact of obesity on lymphocyte function is of particular importance.

TRAF5 has been implicated in regulating the cytokine network associated with obesity. *Traf5*-deficient (*Traf5^−/−^*) mice fed a high-fat diet (HFD; 45 kcal% or 60 kcal% from fat) exhibit elevated levels of pro-inflammatory cytokines, such as TNF-α, in the liver, adipose tissue, and serum, compared with those fed a normal diet (ND; 10 kcal% from fat) [26,28]. *Traf5* deficiency reduces CD40-mediated antibody responses [14,21], while enhancing interleukin-6 (IL-6)- and IL-27-induced differentiation of Th17 and Th1 cells, respectively [20,24]. Members of TRAF family, including TRAF5, differentially contribute to atherogenesis by regulating CD40 signaling in endothelial cells [64]. The CD40L–CD40–TRAF signaling cascade plays a pivotal role in atherosclerosis by initiating a wide range of inflammatory processes, which can be either detrimental or beneficial depending on the context [65].

*Traf5^−/−^* mice display exacerbated obesity-related disease phenotypes [23,25,26,27,28]. These mice develop more severe adipose tissue inflammation, insulin resistance, and hepatic steatosis when challenged with an HFD. They also exhibit accelerated atherosclerosis characterized by larger, lipid-rich plaques, along with aggravated cardiac hypertrophy, fibrosis, and myocardial injury under stress conditions. Across these models, the absence of *Traf5* leads to heightened inflammatory responses, increased immune cell infiltration, and more pronounced metabolic and cardiovascular dysfunction.

In humans, reduced TRAF5 expression has been observed in various obesity-related conditions, underscoring its clinical relevance. In adipose tissue, TRAF5 levels are lower in obese individuals but can be restored following bariatric surgery [28]. In the liver, patients with nonalchoholic fatty liver disease also show reduced TRAF5 expression, consistent with findings in obese mouse models where Traf5 deficiency exacerbates steatosis and inflammation [26]. Furthermore, individuals with stable or acute coronary artery disease exhibit significantly reduced TRAF5 mRNA levels in blood compared to healthy controls [23], aligning with the accelerated atherosclerosis phenotype observed in *Traf5^−/−^* mice. These findings suggest that TRAF5 downregulation in humans is associated with obesity, fatty liver, and cardiovascular disease, and that loss of TRAF5 function may contribute to the increased inflammation and tissue damage underlying cardiometabolic diseases. Despite these insights, the mechanisms by which TRAF5 regulates immune responses under HFD conditions remain poorly understood. In particular, the role of TRAF5 in maintaining the functional integrity of B and T lymphocytes and their associated antibody responses in animals with diet-induced obesity has not been clearly defined.

Therefore, in this study, we characterized the phenotypes of B and T cells in the spleens of HFD-fed wild-type (*Traf5^+/+^*) and *Traf5^−/−^* mice. Consistent with previous reports [26,28], levels of the pro-inflammatory cytokine TNF-α were significantly elevated in the spleens of *Traf5^−/−^* mice. Notably, although the overall populations of B and T cells in the spleen were broadly comparable between the two groups, the number of immunoglobulin G2c (IgG2c)-producing cells was significantly reduced in *Traf5^−/−^* mice compared to *Traf5^+/+^* controls, despite increased expression of CD40L in the spleens of *Traf5^−/−^* mice. Furthermore, CD40-mediated class switching to IgG2c was severely impaired in *Traf5^−/−^* B cells. These findings demonstrate, for the first time, that reduced TRAF5 expression impairs CD40-dependent humoral immune responses under HFD conditions, potentially contributing to the development of cardiovascular and autoimmune diseases.

## 2. Result

### 2.1. Elavated Splenic Expression of the Pro-Inflammatory Cytokine mRNAs Tnf (TNF-α) and Cd40lg (CD40L) Is Observed in Traf5^−/−^ Mice Fed a High-Fat Diet

Elevated levels of pro-inflammatory cytokines have been reported in hepatocytes, adipocytes, and blood of *Traf5^−/−^* mice fed an HFD [26,28]. In this study, to determine whether these cytokines are also elevated in the spleen, we assessed the gene expression levels of pro-inflammatory mediators in the spleens of *Traf5^+/+^* (WT) and *Traf5^−/−^* (KO) mice after 27 weeks of HFD treatment (Figure 1A). At the start of the treatment, the body weights of WT and KO mice were comparable (WT, *n =* 6: 24.7 ± 0.8 g; KO, *n* = 8: 24.5 ± 0.4 g; *p* = 0.874, Student’s *t*-test). At the end of the treatment, no significant difference in body weight was observed between the groups (WT, *n* = 6: 55.4 ± 1.5 g; KO, *n* = 8: 54.8 ± 0.8 g; *p* = 0.714, Student’s *t*-test). A trend toward impaired glucose tolerance was observed in HFD-fed KO mice compared with HFD-fed WT mice, although the difference was not statistically significant (Appendix A).

After 27 weeks of HFD feeding, splenocytes from KO mice exhibited significantly increased expression of TNF superfamily members TNF-α (encoded by *Tnf*) and CD40L (encoded by *Cd40lg*), with levels approximately twofold higher than those observed in WT mice under HFD-induced inflammatory conditions (Figure 1B). In contrast, the mRNA expression levels of other cytokine signaling-related genes, including *Ifng* (IFN-γ), *Il4* (IL-4), *Il6* (IL-6), *Il10* (IL-10), *Il17f* (IL-17F), *Il21* (IL-21), *Il23r* (IL-23R), *Il27* (IL-27A)*, Ebi3* (IL-27B), and *Socs3* (SOCS3), were comparable between HFD-fed WT and KO mice (Appendix A). As a control, we confirmed that the expression levels of *Tnf* and *Cd40lg* did not significantly differ between WT and KO mice fed a normal diet (ND) for the same duration (Figure 1C). These results suggest that the loss of TRAF5 expression selectively upregulates a subset of pro-inflammatory cytokines in the spleen under HFD conditions.

### 2.2. IgG2c-Producing Cells Are Decreased in the Spleens of Traf5^−/−^ Mice Fed a High-Fat Diet

We previously demonstrated that TRAF5 plays a critical role in CD40 signaling in B cells and humoral immune responses [21]. Although CD40L expression was upregulated in the spleen (Figure 1B), we hypothesized that CD40 signaling might be impaired in lymphocytes, particularly in B cells, from HFD-fed KO mice. Specifically, we aimed to determine whether *Traf5* deficiency in B cells compromises CD40-mediated antibody production.

Serum IgM and IgG2c antibody levels were comparable between HFD-fed WT and KO mice (Appendix A). The total number of splenocytes was also similar between the two groups (Appendix A). Furthermore, the splenic B cell populations did not significantly differ in either the percentage or the absolute number of total B220^+^ B cells, follicular B cells (B220^+^CD23^high^CD1d^low^), or marginal zone B cells (B220^+^CD23^low^CD1d^high^) (Appendix A).

Surprisingly, the number of IgG2c-producing cells was significantly lower in the spleens of HFD-fed KO mice than in those of HFD-fed WT mice—approximately a 1.5-fold reduction (Figure 2A,C). However, no significant differences were observed in the populations of IgM-, IgG1-, IgG3-, or IgA-producing cells between the groups (Figure 2A and Appendix A). In contrast, the numbers of these antibody-producing cell populations, including IgG2c-producing cells, were comparable between ND-fed WT and KO mice, serving as a control (Figure 2B,C and Appendix A
). These findings indicate that, although the difference is modest, TRAF5 contributes to the production of a restricted subclass of IgG in the spleens of HFD-fed mice and suggest that CD40-mediated signaling is impaired in B cells from HFD-fed KO mice.

### 2.3. Splenic T Cell Phenotype in High-Fat Diet-Fed Traf5^+/+^ and Traf5^−/−^ Mice

In metabolic diseases, the balance between Th17 and Tregs is regulated by inflammatory cytokines and metabolic factors, including fatty acids and other diet-derived molecules [42,66]. TRAF5 supports signaling through members of the TNFR superfamily expressed on T cells [12,14,17,20,67]. In contrast, TRAF5 inhibits STAT3 signaling, which is mediated by the common gp130 receptor for IL-6, thereby restraining the differentiation of naïve CD4^+^ T cells into Th17 cells [24]. Therefore, *Traf5* deficiency may disrupt T cell homeostasis in the spleen under HFD conditions.

Flow cytometric analysis revealed that the percentages of CD4^+^, CD8^+^, CD4^+^CD44^high^CD62L^low^ (effector memory), and CD8^+^CD44^high^CD62L^low^ (effector memory) T cell populations were significantly lower in the spleens of HFD-fed KO mice compared to HFD-fed WT mice (Appendix A). Although the absolute numbers of CD4^+^, CD8^+^, and CD4^+^CD44^high^CD62L^low^ T cells tended to decrease, the differences were not statistically significant between the groups. In contrast, the absolute number of CD8^+^CD44^high^CD62L^low^ T cells was significantly reduced in HFD-fed KO mice—approximately a 1.6-fold reduction (Appendix A). Both the percentages and absolute numbers of CD4^+^CD44^low^CD62L^high^ (naïve), CD8^+^CD44^low^CD62L^high^ (naïve), and CD8^+^CD44^high^CD62L^high^ (central memory) T cell populations were comparable between the groups (Appendix A). These findings suggest that TRAF5 expression regulates peripheral T cell homeostasis under HFD conditions, particularly in CD8^+^CD44^high^CD62L^low^ T cells. Based on these results, it is plausible that TRAF5 contributes to the maintenance of effector and memory T cells in the periphery, possibly through modulation of survival signaling pathways.

Next, we examined lineage-specific transcription factors for effector CD4^+^ T cells, including T-bet (Th1), Gata3 (Th2), RORγt (Th17), Foxp3 (Treg), and Bcl6 (T follicular helper, Tfh), as well as cytokines associated with these effector CD4^+^ T cell subsets. The Th17-related transcription factor *Rorc* (encoding RORγt) was significantly upregulated in the spleens of HFD-fed KO mice compared to HFD-fed WT mice (Appendix A), whereas ND-fed WT and KO mice showed similar expression levels (Appendix A). In contrast, the expression of other transcription factors, including *Tbx21* (T-bet), *Gata3*, *Foxp3*, and *Bcl6*, did not differ significantly between the groups (Appendix A). Furthermore, the expression levels of CD4^+^ T cell-associated cytokines, such as *Ifng*, *Il4*, *Il10*, and *Il21*, were also not significantly different between the groups (Appendix A). Similarly, the expression of Th17- and Tfh-related genes, including *Il6*, *Il17f*, *Il23r*, *Il27*, *Ebi3*, and *Socs3*, was comparable between the groups (Appendix A). These findings suggest that, although the underlying mechanisms remain unclear, *Traf5* deficiency may contribute to the upregulation of Th17-type responses in the spleens of HFD-fed mice.

### 2.4. TRAF5 Promotes Aicda (AID) Expression and Class-Switch Recombination to IgG2c Mediated by Soluble CD40L and IFN-γ in B Cells from Normal-Diet-Fed Mice

Previous studies have shown that TRAF5 supports B cell proliferation, the expression of CD40-target genes, and the production of antibodies mediated by CD40 under normal dietary conditions [14,21]. However, it remains unclear whether TRAF5 is involved in regulating class-switch recombination (CSR), which is mediated by activation-induced cytidine deaminase (*Aicda*, also known as AID) [68]. To investigate this, we examined whether IgG2c production is impaired in B cells from ND-fed KO mice in vitro. IgG2c production can be induced by co-stimulation with CD40 and IFN-γ signaling [69]. Splenic B cells from both groups were cultured for six days in the presence or absence of soluble CD40L (recombinant Fc–CD40L protein [21]) and IFN-γ. Notably, under these stimulation conditions, B cells from KO mice failed to produce IgG2c, even when both CD40L and IFN-γ were presented—showing levels comparable to those observed with CD40L alone, IFN-γ alone, or the unstimulated control. In contrast, B cells from WT mice produced significantly higher levels of IgG2c only when co-stimulated with both CD40L and IFN-γ, with IgG2c levels in KO B cells approximately 6.6-fold lower than those in WT B cells (Figure 3A). Importantly, CD40-mediated upregulation of *Aicda* (AID) was markedly impaired in KO B cells, showing approximately a 4.4-fold reduction compared to WT B cells (Figure 3B). These findings demonstrate that TRAF5 regulates *Aicda* expression through CD40 signaling and suggest that the reduced IgG2c production observed in obese KO mice is due to defective CSR mediated by CD40 signaling.

### 2.5. CD40-Mediated IgG2c Production Is Reduced in Splenocytes from Traf5^−/−^ Mice Fed a High-Fat Diet

Although *Ifng* mRNA expression in the spleen was comparable between HFD-fed WT and KO mice (Appendix A), we investigated whether *Traf5* deficiency impairs CD40-mediated IgG2c production in splenocytes from HFD-fed mice. Whole splenocytes from HFD-fed WT and KO mice were stimulated with soluble CD40L for four days, and IgG2c levels in the culture supernatants were quantified by ELISA. Consistent with previous findings (Figure 3), CD40L stimulation significantly enhanced IgG2c production in splenocytes from HFD-fed WT mice, even in the absence of IFN-γ. In contrast, the same condition failed to elicit a productive response in splenocytes from HFD-fed KO mice, resulting in an approximately 1.9-fold reduction compared to WT mice (Figure 4A). In contrast, CD40L stimulation failed to induce IgG2c production in splenocytes from ND-fed WT or KO mice, which served as controls (Figure 4B). These results highlight a critical role for TRAF5 in promoting CD40-driven CSR, particularly under inflammatory conditions associated with HFD.

## 3. Discussion

In this study, we investigated the role of TRAF5 in regulating immune responses—particularly those mediated by splenic B and T cells—under conditions of obesity-induced inflammation. Our results demonstrated that *Traf5^−/−^* mice fed an HFD exhibited a significant reduction in IgG2c-producing cells in the spleen, despite elevated expression of the pro-inflammatory cytokines TNF-α and CD40L. Although the splenic B cell population was comparable to that of wild-type controls, in vitro assays revealed that the impaired IgG2c production was due to defective CD40–TRAF5 signaling. These findings identify TRAF5 as a mediator of CD40-driven antibody class-switch recombination during chronic inflammation, highlighting its potential role in obesity-associated immune dysregulation.

The expression levels of *Tnf* and *Cd40lg* mRNAs were significantly elevated in the spleens of *Traf5^−/−^* mice. Previous studies have reported increased levels of pro-inflammatory cytokines in the blood, visceral fat, liver, and adipose tissue of *Traf5^−/−^* mice [26,28]. Although our findings are consistent with these reports, no previous study has specifically examined inflammatory responses in a secondary lymphoid organ, such as the spleen, under HFD conditions. Furthermore, in mice fed an ND, *Tnf* and *Cd40lg* expression levels did not differ significantly between *Traf5^+/+^* and *Traf5^−/−^* mice, indicating that the observed upregulation is specific to HFD conditions. This raises the question of how this selective upregulation is regulated.

TRAF5 deficiency is associated with the upregulation of pro-inflammatory mediators in injured tissues and activated immune cells. Cardiac hypertrophy induced by transthoracic aorta construction increases the expression of pro-inflammatory cytokines, including TNF-α, in the cardiac tissues of *Traf5^−/−^* mice [25]. Similarly, skin wound healing promotes the induction of pro-inflammatory cytokines, including TNF-α, in *Traf5^−/−^* mice [70]. During myocardial reperfusion injury, *Traf5* deficiency exacerbates inflammatory responses in cardiac tissue and further enhances the induction of pro-inflammatory cytokines, including TNF-α [27]. *Traf5^−/−^* B cells produce higher levels of pro-inflammatory cytokines, including TNF-α, in response to ligands for TLR4, TLR7, and TLR9 [71]. Likewise, upon stimulation with TLR7 or TLR9 ligands, *Traf5^−/−^* plasmacytoid dendritic cells secrete increased amounts of pro-inflammatory cytokines, including TNF-α [70]. A possible explanation for the upregulation of the TNF superfamily members TNF-α and CD40L in the spleen is that splenic immune cells elevate *Tnf* and *Cd40lg* expression in response to metabolic stressors such as oxidized low-density lipoprotein, fatty acids, and self-DNA/RNA from dying or necrotic cells [48,72,73,74,75], under the metabolically abnormal conditions induced by HFD feeding. These TLR-mediated mechanisms warrant further investigation in future studies.

In *Traf5^−/−^* mice fed an HFD, the number of IgG2c-producing cells in the spleen was significantly reduced compared to *Traf5^+/+^* mice fed an HFD. The CD40L-CD40 interaction provides a critical signal for B cell proliferation, CSR, and antibody production [76,77,78,79]. CD40 contains two TRAF-binding domains—TRAF6 and TRAF2/3/5—located in its cytoplasmic tail, with each TRAF binding to its respective domain to regulate downstream signaling [65]. Previous studies have shown that *Traf5^−/−^* B cells exhibit reduced proliferation and diminished expression of CD40 target genes, such as CD86, CD80, and CD23 [7,14,21]. Furthermore, TRAF5 acts downstream of CD40 to promote T cell-dependent humoral responses in vivo [21]. In this study, CD40L-stimulated splenocytes from HFD-fed *Traf5^−/−^* mice showed significantly impaired IgG2c production. Additionally, CD40-stimulated B cells derived from *Traf5^−/−^* mice fed an ND exhibited reduced expression of *Aicda* (AID), a gene essential for CSR. These findings suggest that splenic B cells in HFD-fed *Traf5^−/−^* mice have attenuated CD40 signaling, resulting in impaired IgG2c class switching. Notably, B cells lacking TRAF2—but not TRAF3—display defective activation of the NF-κB1 complex via CD40, leading to suppressed AID expression and reduced antibody responses in ND-fed mice [80]. TRAF5 is structurally and functionally related to TRAF2 [5,16,81,82]. Whether TRAF5 regulates CSR through a mechanism similar to that of TRAF2 remains to be elucidated.

Both CD40 and IFN-γ signaling pathways are essential for IgG2c class switching [69]. Although IFN-γ production is typically elevated in the adipose tissue of obese mice [83], no significant differences in *Ifng* mRNA levels were observed in the spleens of HFD-fed *Traf5^+/+^* and *Traf5^−/−^* mice in this study. Similarly, the expression of IFN-γ receptor target genes, including *Tbx21* (T-bet), *Stat1*, and *Irf1*, was comparable between the two genotypes of CD19^+^ B cells. Furthermore, purified CD19^+^ B cells from HFD-fed *Traf5^+/+^* and *Traf5^−/−^* mice failed to induce IgG2c in response to CD40L stimulation, suggesting that additional factors within the splenic microenvironment are necessary for effective in vivo IgG2c induction. Together, these findings suggest that impaired CD40 signaling in *Traf5^−/−^* B cells under HFD conditions underlies the reduction in CSR, thereby resulting in diminishing IgG production. Further studies are warranted to elucidate the underlying mechanisms and to clarify their potential relevance to the development of autoimmunity.

IgG antibody subclasses mediate distinct effector functions in host defense against microbial pathogens [84,85]; therefore, reduced production of a specific subclass may compromise the immune system’s capacity to combat infection. Our findings suggest that *Traf5* deficiency limits IgG2c production under conditions of mild, chronic inflammation associated with obesity. These results imply that reduced TRAF5 expression may not only exacerbate obesity-related metabolic disturbances but could also predispose obese individuals to infection-related complications. Further studies will be important to clarify the role of TRAF5 in regulating B cell-mediated humoral immunity under both physiological and pathological conditions, thereby advancing our understanding of its contribution to maintaining robust adaptive immunity.

A notable limitation of this study is the relatively small and heterogenous number of mice across experimental groups, primarily due to breeding inefficiencies and attrition over the experimental timeline. Specifically, *Traf5*^+/−^ intercrosses produced male offspring with the desired genotypes at inconsistent ratios, preventing uniform group allocation. In addition, several animals were excluded during long-term maintenance because of health deterioration or facility-related complications, and a small number of samples were lost during data acquisition. Although all available littermates were included to reduce inter-group variability, these constraints inevitably resulted in imbalanced cohort sizes. This study is exploratory in nature, and these limitations should be considered when interpreting the findings.

Another limitation is the absence of *Traf5*^+/−^ (heterozygous) mice as a control group. While *Traf5^+/+^* (WT) and *Traf5^−/−^* (KO) littermates were used to minimize genetic variability, incorporating heterozygous mice could provide insight into gene dosage effects and potential polymorphisms. Further studies will therefore consider including *Traf5*^+/−^ mice to strengthen the experimental design.

Additionally, we did not measure serum cholesterol or lipid profiles in this study. This decision reflects the primary focus of our investigation, which was to assess the functionality of B and T lymphocytes in the spleen of HFD-fed *Traf5^+/+^* and *Traf5^−/−^* mice, rather than to comprehensively characterize systemic metabolic parameters. Future studies incorporating detailed metabolic profiling will be necessary to fully elucidate the relationship between Traf5 deficiency, immune function, and systemic lipid metabolism.

Although rigorous statistical analyses were employed, the restricted sample sizes may limit the robustness of the conclusions, warranting cautious interpretation. Future investigations with larger and more evenly distributed cohorts will be essential to confirm and extend the present findings. Nonetheless, the reproducible trends observed across independent experiments underscore the biological relevance of the results presented herein.

In conclusion, our findings reveal that diminished TRAF5 expression, within the milieu of obesity-associated low-grade chronic inflammation, compromises CD40L-CD40-TRAF5-meidated humoral immunity, thereby potentially contributing to the pathogenesis of cardiovascular and autoimmune disorders. As a central signaling adaptor for members of the TNFR superfamily, TRAF5 appears to be critical for sustaining adaptive immune responses, offering mechanistic insights into immune dysregulation in obesity-related inflammatory conditions and underscoring its potential as a therapeutic target.

## 4. Materials and Methods

### 4.1. Mice

Heterozygous *Traf5^+/−^* mice on a C57BL/6 background were intercrossed to generate *Traf5^+/+^* and *Traf5^−/−^* mice [14,24]. In this study, only male mice (3–8 per experiment) were included. They were fed either a 60 kcal% HFD (D12492, Research Diets Inc., New Brunswick, NJ, USA) or a standard chow diet (normal diet (ND); 12 kcal%, CLEA Rodent Diet CE-2, CLEA Japan Inc., Tokyo, Japan), starting at 6–7 weeks of age. After 27 weeks of HFD feeding and a 4 h fast, the mice were euthanized by cervical dislocation. The mice were bred and housed in a temperature-controlled environment (24 ± 3 °C) with a 12 h light/dark cycle (lights on at 7:00 AM) and had ad libitum access to food and water under specific pathogen-free conditions at the Life Science Research Center, University of Toyama. Animal experimental protocols were approved by the Animal Care and Use Committee of the University of Toyama (Approval Numbers: A2024PHA-02 and A2024PHA-03) and were conducted in accordance with the Institutional Animal Experiment Handling Rules of the University of Toyama.

### 4.2. Antibodies

The antibodies used in this study are listed in Appendix A.

### 4.3. ELISPOT Assay

The enzyme-linked immunospot (ELISPOT) assay was performed to detect antibody-producing cells of each isotype in the spleen, as previously described [21,86]. Briefly, splenocytes from mice fed either an HFD or an ND (ranging from 5 × 10^3^ to 1 × 10^6^ cells/well) were added to ELISPOT plates (MultiScreen HTS^TM^ MSHAS4510, Merck KGaA, Darmstadt, Germany) pre-coated with 10 μg/mL of unlabeled goat anti-mouse Ig (1010-01, Southern Biotech, Birmingham, AL, USA). To detect each antibody isotype, alkaline phosphatase-conjugated goat anti-mouse IgA, IgM, IgG1, IgG2c, or IgG3 was added to the wells, followed by the addition of Nitroblue Tetrazolium (NBT) (022-08661, FUJIFILM Wako, Osaka, Japan) and 5-bromo-4-chloro-3-indolyl phosphate (BCIP) (148-01991, FUJIFILM Wako) as substrates. After imaging each well, the number of spots was counted. The number of IgA-, IgM-, IgG1-, IgG2c-, and IgG3-producing cells per spleen was then calculated by multiplying the total number of splenocytes by the frequency obtained from the ELISPOT assay.

### 4.4. Detection of IgG2c

To induce IgG2c production, single-cell suspensions of splenocytes (5 × 10^5^ cells/well) from *Traf5^+/+^* and *Traf5^−/−^* mice fed either an HFD or an ND were cultured for four days with 10 µg/mL recombinant Fc–CD40L protein [21] in RPMI 1640 medium (189-02025, FUJIFILM Wako), supplemented with 10% heat-inactivated fetal calf serum, 100 U/mL penicillin, 100 µg/mL streptomycin, 2 mM L-alanyl-L-glutamine, and 50 µM 2-mercaptoethanol. Resting B cells (1 × 10^5^ cells/well), isolated from the spleens of *Traf5^+/+^* and *Traf5^−/−^* mice using CD43 (Ly48) microbeads (130-049-801, Miltenyi Biotec, Bergisch Gladbach, Germany), were cultured for six days with 30 µg/mL Fc–CD40L and 6 ng/mL recombinant mouse IFNγ (575302, Biolegend, San Diego, CA, USA) in complete RPMI 1640 medium. The culture supernatant was collected and added to ELISA plates (439454, Thermo Fisher Scientific, Waltham, MA, USA) pre-coated with 0.25 μg/mL unlabeled goat anti-mouse Ig (1010-01, Southern Biotech). To quantify IgG2c in the culture supernatants, alkaline phosphatase-conjugated goat anti-mouse IgG2c (1078-04, Southern Biotech) was used at a 1:1000 dilution, along with the Alkaline Phosphatase Yellow (pNPP) Liquid Substrate System for ELISA (P7998-100ML, Merck, Burlington, MA, USA). Absorbance was measured at 405 nm using a FilterMax F5 microplate reader (Molecular Devices, San Jose, CA, USA). IgG2c antibody titers were determined by comparing the test samples to a standard curve generated using a representative sample assigned an arbitrary unit value of one.

### 4.5. Real-Time RT-PCR

Total RNA was extracted from mouse spleens or from resting B cells stimulated with 30 μg/mL Fc–CD40L on day 2, using either ISOGEN II (Nippon Gene, Toyama, Japan) or TRI Reagent (TR118, Molecular Research Center, Inc., Cincinnati, OH, USA). Complementary DNA (cDNA) was synthesized using ReverTra Ace^®^ qPCR RT Master Mix with gDNA Remover (FSQ-301, TOYOBO, Osaka, Japan). The diluted cDNA was subjected to quantitative real-time RT-PCR using the Brilliant III Ultra-Fast SYBR^®^ qPCR Master Mix (600882, Agilent Technologies, Santa Clara, CA, USA) on a CFX Connect^TM^ Real-Time PCR Detection System (Bio-Rad Laboratories, Inc., Hercules, CA, USA). The primers used in this study are listed in Appendix A. The expression levels of target genes were normalized to those of *Stx5a* or *Rn18s*. To justify the use of *Stx5a* as a housekeeping gene, we confirmed that the Cq values of three commonly used reference genes—*Stx5a*, *Hprt*, and *Gapdh*—in the spleens of HFD-fed WT and KO mice were comparable, with minimal variation among samples. Given this stability, *Stx5a* was selected as the reference gene for normalization in this experiment. 

### 4.6. Flow Cytometry

Splenocytes were stained with antibodies against cell surface markers as described in Appendix A. Dead cells were excluded by staining with propidium iodide (PI; 341-078881, FUJIFILM Wako). Flow cytometric data were acquired using a Celesta flow cytometer (BD Biosciences, Franklin Lakes, NJ, USA) and analyzed with BD FACSDiva software (version 9).

### 4.7. Statistical Analysis

Statistical significance was assessed using either a two-tailed Student’s *t*-test or the Tukey–Kramer multiple comparison test, performing in JMP Pro (version 18.0.1; SAS Institute, Cary, NC, USA) or in EZR (version 1.36) [87]. All experiments were performed at least twice, yielding similar results unless otherwise specified. A *p*-value < 0.05 was considered statistically significant.

## Figures and Tables

**Figure 1 ijms-26-09494-f001:**
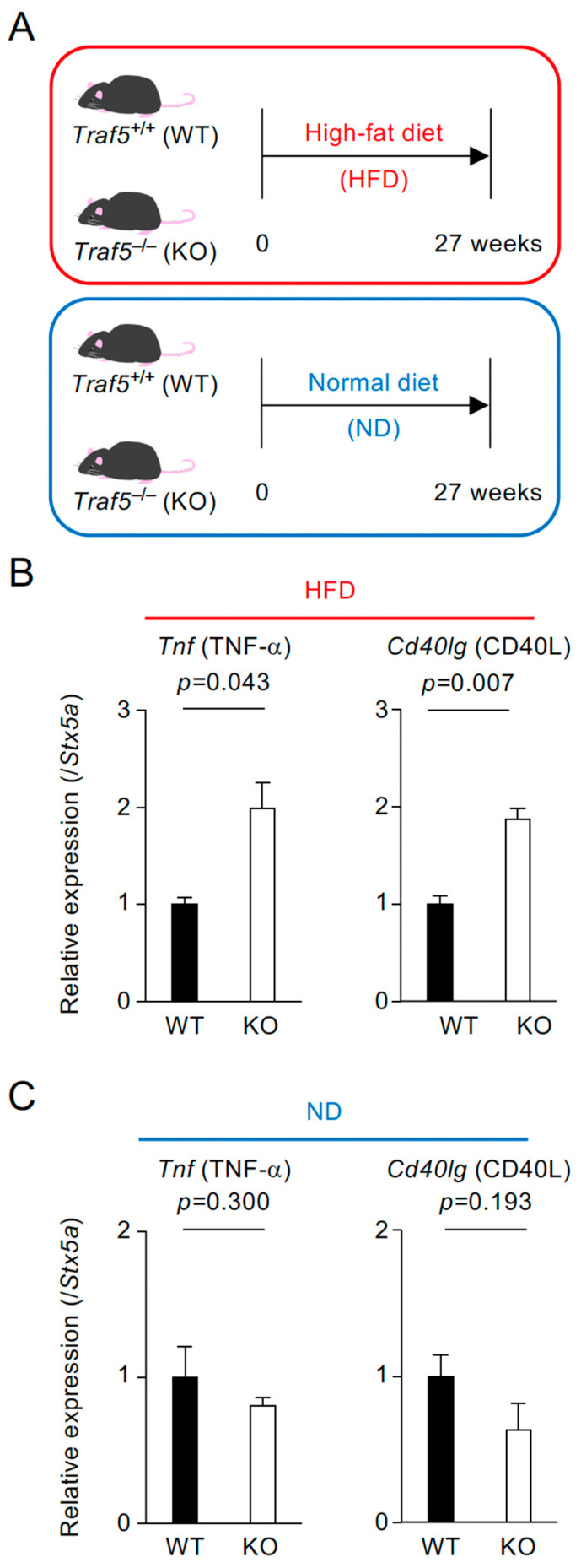
Pro-inflammatory cytokines TNF-α and CD40L are significantly upregulated in the spleens of *Traf5^−/−^* mice fed a high-fat diet. (**A**) Experimental schema. *Traf5^+/+^* (WT) and *Traf5^−/−^* (KO) mice were fed either a high-fat diet (HFD) or a normal diet (ND) ad libitum for 27 weeks. (**B**,**C**) Expression levels of *Tnf* (TNF-α) and *Cd40lg* (CD40L) in splenocytes from WT and KO mice fed an HFD (**B**) or an ND (**C**), as assessed by real-time RT-PCR. Data are presented relative to *Stx5a* expression as the mean ± standard error of the mean ((**B**): WT (*n* = 3), KO (*n* = 3); (**C**): WT (*n* = 4), KO (*n* = 5)). For each gene, the expression level in WT mice was set to 1. *p*-values were determined using Student’s *t*-test.

**Figure 2 ijms-26-09494-f002:**
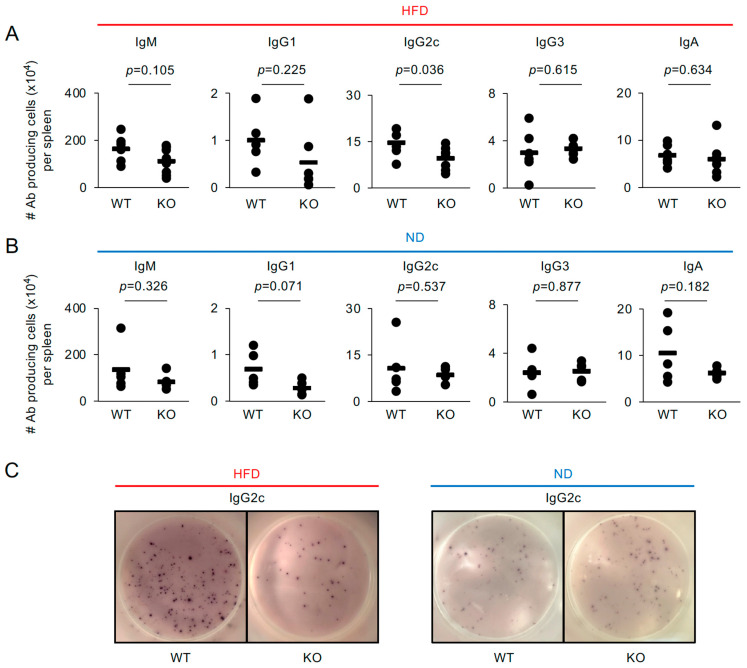
IgG2c-producing cells are significantly decreased in the splenocytes of *Traf5^−/−^* mice fed a high-fat diet. The total numbers of IgM-, IgG1-, IgG2c-, IgG3-, and IgA-producing cells in the spleens of WT and KO mice fed an HFD (**A**,**C**; as shown in Figure 1B) or an ND (**B**,**C**; as shown in Figure 1C) were determined by ELISPOT. (**C**) Representative images of ELISPOT wells showing IgG2c-producing cells, as shown in (**A**,**B**). Bars indicate mean values, and dots represent individual mice ((**A**): IgG1 [WT: *n* = 5, KO: *n* = 7], IgM, IgG2c, IgG3, and IgA [WT: *n* = 6, KO: *n* = 8]; (**B**): all antibodies [WT: *n* = 5, KO: *n* = 5]). *p*-values were determined using Student’s *t*-test.

**Figure 3 ijms-26-09494-f003:**
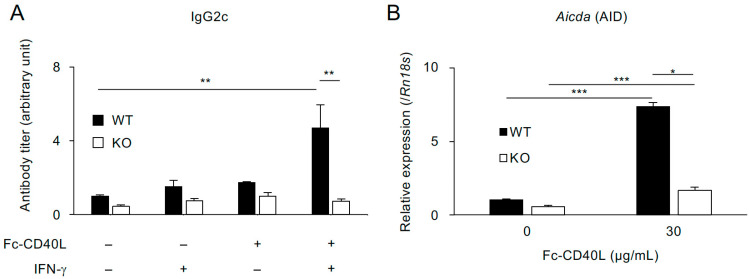
*Aicda* (AID) expression and IgG2c production mediated by soluble CD40L were impaired in splenic B cells from *Traf5^−/−^* mice fed an ND. (**A**) Splenic B cells from WT and KO mice fed an ND at 6–12 weeks of age were cultured for six days with or without Fc–CD40L (30 μg/mL) in the presence or absence of IFN-γ (6 ng/mL). IgG2c levels in the culture supernatants were measured by ELISA. Data are presented as mean ± standard error of the mean (*n* = 3). IgG2c levels in WT B cells without Fc–CD40L and IFN-γ were normalized to 1. (**B**) Expression levels of *Aicda* (AID) mRNA in splenic B cells from WT and KO mice, stimulated with Fc–CD40L (30 μg/mL) for two days as in (**A**), were assessed by real-time RT-PCR. Data are presented relative to *Rn18s* expression as mean ± standard error of the mean (*n* = 3). * *p* < 0.05, ** *p* < 0.01, and *** *p* < 0.001 indicate statistically significant differences among the groups, as determined by the Tukey–Kramer test.

**Figure 4 ijms-26-09494-f004:**
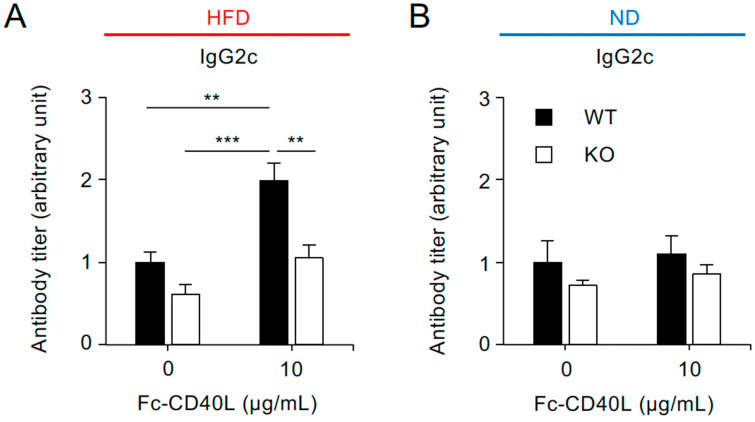
Soluble CD40L stimulation does not enhance IgG2c production in the splenocytes of *Traf5^−/−^* mice fed a high-fat diet. Splenocytes from WT and KO mice fed an HFD ((**A**); as shown in Figure 1B, Figure 2A and Figure 4A) or an ND ((**B**); as shown in Figure 1C, Figure 2B and Figure 4B) were cultured with or without Fc–CD40L (10 μg/mL) for four days. IgG2c levels in the culture supernatants were measured by ELISA. Data are presented as mean ± standard error of the mean ((**A**): WT (*n* = 6), KO (*n* = 8); (**B**): WT (*n* = 6), KO (*n* = 5)). IgG2c levels in WT splenocytes without Fc–CD40L were normalized to 1. ** *p* < 0.01 and *** *p* < 0.001 indicate statistically significant differences among the groups, as determined by the Tukey–Kramer test.

## Data Availability

The original contributions presented in the study are included in the article and Appendix A, further inquiries can be directed to the corresponding author.

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
