# Peer review of "Defective IgG Class Switching in the Spleen of TRAF5-Deficient Mice Reveals a Role for TRAF5 in CD40-Mediated B Cell Responses During Obesity-Associated Inflammation"

_ijms, 2025, doi:10.3390/ijms26199494_

Round 1

Reviewer 1 Report

Comments and Suggestions for Authors

I found the article rather poorly written for a wide audience at the international level of molecular sciences. We need to be aware of IgG2c, Traf5, TRAF5, CD40, B cell, and all too many acronyms that are synonymous to a poorly written article. Please make efforts to gain clarity.

TNF, TRAF1, TRAF7, Cd40lg, CSR, HFD, … abstract does not help to gain clarity. Does it mean that interleukin is less important and has no acronym?

In Introduction, the gap between TRAF and obesity is very sudden, and there is certainly more physiological or biochemical steps to describe before to mention about obesity disease.

For example, the authors start by mentioning about tumor necrosis and immune response, nothing about lipid metabolism, how to end up with obesity which is a pathology related to deficiencies in the lipid metabolism process. This is the deficiency in lipid process which brings the patient at risk, with altered immune system, not the obesity itself.

Paragraph 1: TRAF, Paragraph 2: Obesity and all related deficiencies, Paragraph 3: TRAF….. please better organize the text and gain clarity…..

What is an “obese animal”?

When TRAF is so important in the immune and inflammatory response, I have hard time to believe this is the first time that there is something reported about the link between TRAF and CD40 dependent humoral immune responses. Is it the fact that all these physiological and biochemical parameters are measured in obesity conditions that make them so appealing?

Many reviews and research articles about TRAF, interleukin (IL), atherosclerosis and obesity (in human and rodents). What is so novel here? The authors describe nothing in term of clinical obesity and heart problems.
Most parts of results are introduction. Definition of spleen, etc. Figure 1 that shows the display of the experience is a nice initiative (though WT? and KO? Should be separated), however, the results are rather unclear and not so convincing. The variation of the relative expression levels do not exceed two-fold (see Figure 1). How can it be significant from a physiological point of view? Also, how can it be significant from a statistical point of view? There is a different number of individuals tested in the sampling: n=3, n=4, n=5, …… the number of individuals is too small, and too varying between the WT?  and KO? Groups.

Same remark for figure 2. I am surprised that the authors focus on IgG2c, while there is similar variation in IgG1, though not in the same conditions. In fact, a good interpretation of Figure might be that every immunoglobulin is affected and shows differences between WT? and KO?. What is the main point to focus on Figure 2C that only deals with IgG2c? What are the results for the other immunoglobulins focuses on IgG2c?

The results are rather odd on Figure 3. If we consider the antibody titers in presence of FcCD40L in KO?, we found a decrease in the titers, which probably correspond to interindividual variations, and is not related to any sort of treatment. Since n is only related to 3, I don’t think there is any truly relevant here. What is compared here? When we compare WT and KO in each condition, there is increase of antibody titer in absence and presence of FcCD40L, so how to interpret these data? Same remark for the IFN (+) and (-), titers vary independently of the conditions. There is a marked effect only in IFN(+) and FcCD40(+) conditions. However, if we compare the titers between IFN(+) and FcCD40(+) and IFN(-) and FcCD40(+) conditions, there is only a two-fold difference. So, there is little to conclude here. 

The relative expression of Aicda (??) is somehow more intriguing, though it needs to report a clear dose-effect of FcCD40L to be conclusive. 
On figure 4, I see no differences between WT and KO because whatever difference does not exceed 2-fold ranges. 

The number of animals tested vary between groups, n=6, n=8, and n=5 (almost half), how can it be conclusive? Then the question is how can it be conclusive since n was 3 in the other measures, and go to 8 here.

How many mice were available at the beginning of the experiment? How many were included in the WT group? How many were put in the KO group? How many were used for each experiment? The answer to these questions will provide the clue about how relevant this study is. 

Author Response

Please see the attachment, Point-by-Point Responses to Reviewer#1, 20250908.

Reviewer 2 Report

Comments and Suggestions for Authors

1) The title the must be more limited to what happens in the animal splenocyte model.

2) in general, the manuscript confuses ideas because protein factors are sometimes introduced and are not clearly linked to what was said before or after.

3) you must specify which TNF type

4) TNFR line 16 is the same of line 35?

5) please change glucose intolerance with pre-diabetes

6) when you describe experiment you should present a difference by using soluble CD40 ligand and insoluble CD40 ligand. There is a difference since insoluble form is bound to cell membrane!

7) lines 55-58 the sentence is ......like.....may be....

8) line 61 please provide details abount diet HF and normal

9) line 71 show a reference

10) lines 73-74  you mast do experiments also on a traf5 (+/-) to have a whole control ! Animals could have polymorphisms that affect your results

11) Your experiments lack controls: what happen if cells are stimulated by pre-incubating with TNF-alpha, IFN-gamma and LPS.

12) I did not find how animals are male and female?

13) Lines 250-251 the use of LPS is not clear

14) lines 258-260 lack proofs

15) line 262 is not clear what you are comparing

16) authors need to motivate better because they use the Stx5a gene as housekeeping

Comments on the Quality of English Language

In general, the manuscript confuses ideas because protein factors are sometimes introduced and are not clearly linked to what was said before or after. some sentences would be clearer by reversing the subject. some sentences would be put in a form that suggests a possibility where it instead seems like a certainty.

Author Response

Please see the attachment, Point-by-Point Responses to Reviewer#2, 20250908.

Reviewer 3 Report

Comments and Suggestions for Authors

In the manuscript, the authors highlighted Defective IgG2c Class Switching in Traf5-Deficient Mice Reveals a Role for TRAF5 in CD40-Mediated B Cell Responses During Obesity-Associated Inflammation.
The primary objective of the study was to investigate the phenotypes of splenic B and T cells in wild-type (Traf5+/+) and Traf5-/- mice fed a high-fat diet (HFD).
In this study, the authors aimed to characterize the phenotypes of B and T cells in the spleens of wild-type (Traf5+/+) and Traf5-/- mice fed a high-fat diet. Consistent with previous reports, levels of the proinflammatory cytokine TNF-Ρ were significantly elevated in the spleens of Traf5-/- mice. Notably, although the total splenic B and T cell populations were comparable between the two groups, the number of IgG2c-producing cells was significantly reduced in Traf5-/- mice compared to Traf5+/+ controls, despite increased CD40L mRNA expression in the spleens of Traf5-/- mice. Moreover, CD40-mediated class switching to IgG2c was severely impaired in Traf5-/- B cells. These results demonstrate for the first time that downregulation of TRAF5 expression impairs CD40-dependent humoral immune response under lipid-deficient conditions, potentially contributing to the development of cardiovascular and autoimmune diseases. This demonstrates the originality and relevance of the study topic.
These results will expand scientific knowledge and suggest the following: TRAF5 may play a key role in inflammatory conditions characterized by abundant production of TNF family molecules, mediating signaling from its associated TNFRs to the nucleus to regulate lymphocyte activity necessary for adaptive immunity.
The purpose and objectives of the study were fully achieved. The study was conducted at a high scientific and methodological level.
The materials and methods correspond to the purpose and objectives of the study.
The obtained data have been statistically processed.
The text of the manuscript has a logical structure. The text is written in understandable English. There are a small number of spelling and typographical errors in the text.
The results are analyzed in detail, carefully and compared with the available literary sources.
The authors' conclusions correspond to the presented results, arguments and evidence. They fully answer the questions posed.
The conclusion is consistent with the results obtained.
The authors do not allow the use of literary data without reference to them. The list of references corresponds to the problem posed and allows revealing the research questions.
However, the following comments are possible. 1. Abstract. When describing the results of the study, write specific values ​​of the parameter changes. By how much a specific indicator is increased or decreased.
2. Manuscript structure. Should be brought into compliance with the requirements. Materials and methods follow the introduction.
3. Conclusion. Little is presented about the assessment of the data obtained. There are no specific results.
Correction of comments will improve the quality of the manuscript and increase the interest of readers in it.

Author Response

Please see the attachment, Point-by-Point Responses to Reviewer#3, 20250908.

Round 2

Reviewer 1 Report

Comments and Suggestions for Authors

I take note of the effort in improving the quality of the paper. However, key points of criticism remain, to my opinion. The main problems remain, that is the number of individuals tested, the differences between groups (in terms of individuals tested and validity of statistical test), and the 2-fold variation which is described as a strong physiological effect. 

"In this study, only male mice (3–8 per experiment) were 133 included. They were fed either a 60 kcal% HFD (D12492, Research Diets Inc., New Bruns- 134 wick, NJ, USA) or a standard chow diet (normal diet (ND); 12 kcal%, CLEA Rodent Diet 135 CE-2, CLEA Japan Inc., Tokyo, Japan), starting at 6-7 weeks of age."

  • Were only 8 rats used in the whole study? How did you divide such a small group of rats for a study which aim to be conclusive. 3 rats in  a category, even if there is only 3 rats in the other group, this is just not enough in particular if this tackles a first-time experiment, as claimed by the authors.

  • What is there in the diet that make them fat or obese? What is their weight and blood biochemical composition? This information would have been relevant Table. So far tables 1 & 2 are supplementary materials (antibody manufactures and PCR primers, not as important as the cholesterolemia of the rats)

  • In all figures, the standard deviation and the average values do not reflect on strong differences between groups. Relative gene expression does not exceed 2. How can this mean high expression. 

Reviewer 2 Report

Comments and Suggestions for Authors

Please shortly justify in the materials and methods the use of Stx5a as a housekeeping gene as you described:
we evaluated the expression levels of three commonly used reference genes—Stx5a, Hprt, and Gapdh—in the spleens of HFD-fed WT and KO mice using qRT-PCR. Under these conditions, the Cq values of all three genes were comparable, with minimal variation observed between samples (see figure below). Based on this stability, we selected Stx5a as the reference gene for normalization in this experiment. 

Round 3

Reviewer 1 Report

Comments and Suggestions for Authors

I suggested to remove Tables 1 & 2. Since these tables have not been moved to supplementary section, I wonder if all the critical points have been considered.

My understanding of the outcome of this manuscript has not changed: too many limitations for a full-paper at IJMS. It should be revised for a short note or a submission to another journal. 

We have now revised the Discussion section to emphasize that this study is exploratory in nature and that future studies with larger sampling are warranted to confirm and expand these findings (Lines 483-509).

Preliminary "non-conclusive" study should be published in another journal than IJMS, to my understanding.

These findings provide additional insight into glucose metabolism, which is closely related to metabolic health and obesity-associated conditions. However, we did not measure serum cholesterol or lipid profiles in this study, which we acknowledge as a limitation and have noted in the Discussion section (Lines 498-503). 

Yes, there are many biochemical issues in obesity, one of them is key point, cholesterol and lipid metabolism. Glucose is more tuned to diabetes, which confuses the aim of the study. This gives another limitation for publication at IJMS, which needs more clear, complete and conclusive study for publication, to my understanding.

In obesity conditions, every "problematic" parameter is multiplied by 100 It is a complex, multifactorial, still largely preventable disease , affecting individuals of all ages, along with overweight. I would consider a gene expression fluctuating to two-fold up-regulation in overweight conditions, but certainly not in the obese conditions, where the biochemical and physiological problems are extreme. 

"The primary focus of our study was to investigate the functionality of B and T lymphocytes in the spleen of HFD-fed Traf5-deficient mice, rather than to comprehensively characterize systemic metabolic parameters." This study is not conclusive in this matter.
